# Measuring Bayesian Robustness Using Rényi Divergence

**Luai Al-Labadi** [1,*]**, Forough Fazeli Asl** [2]  **and Ce Wang** [1]

1 Department of Mathematical & Computational Sciences, University of Toronto Mississauga, Mississauga, ON L5L 1C6, Canada; ce.wang@mail.utoronto.ca
2 Department of Mathematical Sciences, Isfahan University of Technology, Isfahan 84156-83111, Iran; forough.fazeli@math.iut.ac.ir
* Correspondence: luai.allabadi@utoronto.ca

**Abstract:** This paper deals with measuring the Bayesian robustness of classes of contaminated priors. Two different classes of priors in the neighborhood of the elicited prior are considered. The first one is the well-known $\epsilon$-contaminated class, while the second one is the geometric mixing class. The proposed measure of robustness is based on computing the curvature of Rényi divergence between posterior distributions. Examples are used to illustrate the results by using simulated and real data sets.

**Keywords:** Bayesian robustness; $\epsilon$-contamination; geometric contamination; Rényi divergence



## 1. Introduction

Bayesian inferences require the specification of a prior, which contains a priori knowledge about the parameter(s). If the selected prior, for instance, is flawed, this may yield erroneous inferences.

The goal of this paper is to measure the sensitivity of inferences to a chosen prior (known as *robustness*). Since, in most cases, it becomes very challenging to come up with only a sole prior distribution, we consider a class, $\Gamma$, of all possible priors over the parameter space. To construct $\Gamma$, a preliminary prior $\pi_0$ is elicited. Then robustness for all priors $\pi$ in a neighborhood of $\pi_0$ is intended. A commonly accepted way to construct neighborhoods around $\pi_0$ is through contamination. Specifically, we will consider two different classes of contaminated or mixture of priors, which are given by

$$\Gamma_a = \left\{ \pi(\theta) : \pi(\theta) = (1 - \epsilon)\pi_0(\theta) + \epsilon q(\theta), q \in Q \right\} \tag{1}$$

and

$$\Gamma_g = \left\{ \pi(\theta) : \pi(\theta) = c(\epsilon)\pi_0^{1-\epsilon}(\theta)q^{\epsilon}(\theta), q \in Q \right\}, \tag{2}$$

where $\pi_0$ is the elicited prior, $Q$ is a class of distributions, $c(\epsilon)$ is normalizing constant and $0 \leq \epsilon \leq 1$ is a small given number denoting the amount of contamination. For other possible classes of priors, see for instance, De Robertis and Hartigan (1981) [1] and Das Gupta and Studden (1988a, 1988b) [2,3].

The class (1) is known as the $\epsilon$-contaminated class of priors. Many papers about the class (1) are found in the literature. For instance, Berger (1984, 1990) [4,5], Berger and Berliner (1986) [6], and Sivaganesan and Berger (1989) [7] used various choices of $Q$. Wasserman (1989) [8] used (1) to study robustness of likelihood regions. Dey and Birmiwal (1994) [9] studied robustness based on the curvature. Al-Labadi and Evans (2017) [10] studied robustness of relative belief ratios (Evans, 2015 [11]) under class (1).

On the other hand, the class (2) will be referred as geometric contamination or mixture class. This class was first studied, in the context of Bayesian Robustness, by Gelfand and Dey (1991) [12], where the posterior robustness was measured using Kullback-Leibler

divergence. Dey and Birmiwal (1994) [9] generalized the results of Gelfand and Dey (1991) [12] under (1) and (2) by using the $\phi$ divergence defined by

$$d_\phi(\pi(\theta|x), \pi_0(\theta|x)) = \int \pi_0(\theta|x)\phi(\pi(\theta|x)/\pi_0(\theta|x))\,d\theta \tag{3}$$

for a smooth convex function $\phi$. For example, $\phi(x) = x \ln x$ gives Kullbak-Leibler divergence.

In this paper, we extend the results of Gelfand and Dey (1991) [12] and Dey and Birmiwal (1994) [9] by applying Rényi divergence on both classes (1) and (2). This will give local sensitivity analysis on the effect of small perturbation to the prior. Rényi entropy, developed by Hungarian mathematician Alfréd Rényi in 1961, generalizes the Shannon entropy and includes other entropy measures as special cases. It finds applications, for instance, in statistics [13], pattern recognition [14], economics [15] and biomedicine [16].

Although the focus of this paper is on Rényi divergence, it also contains $(h, \phi)$ family of divergence measures (Menéndez et al., 1995 [17]). Examples of $(h, \phi)$ divergence include Rényi divergence, Shama-Mittal divergence and Bhattacharyya divergence. We refer the reader to Pardo (2006) [18] for more details about $(h, \phi)$ divergence.

An outline of this paper is as follows. In Section 2, we give definitions, notations and some properties of Rényi divergence. In Section 3, we develop curvature formulas for measuring robustness based on Rényi divergence and $(h, \phi)$ divergence. In Section 4, three examples are studied to illustrate the results numerically. Section 5 ends with a brief summary of the results.

## 2. Definitions and Notations

Suppose we have a statistical model that is given by the density function $f_\theta(x)$ (with respect to some measure), where $\theta$ is an unknown parameter that belongs to the parameter space $\Theta$. Let $\pi(\theta)$ be the prior distribution of $\theta$. After observing the data $x$, by Bayes' theorem, the posterior distribution of $\theta$ is given by the density

$$\pi(\theta|x) = \frac{f_\theta(x)\pi(\theta)}{m(x|\pi)},$$

where

$$m(x|\pi) = \int f_\theta(x)\pi(\theta)d\theta$$

is the prior predictive density of the data.

To measure the divergence between two posterior distributions, we consider Rényi divergence (Rényi, 1961 [19]). Rényi divergence of order $a$ between two posterior densities $\pi(\theta|x)$ and $\pi_0(\theta|x)$ is defined as:

$$\begin{aligned} d = d(\pi(\theta|x), \pi_0(\theta|x)) &= \frac{1}{a-1}\ln\left(\int (\pi(\theta|x))^a (\pi_0(\theta|x))^{1-a}d\theta\right) \\ &= \frac{1}{a-1}\ln\left(E_{\pi_0(\theta|x)}\left[\left(\frac{\pi(\theta|x)}{\pi_0(\theta|x)}\right)^a\right]\right), \end{aligned} \tag{4}$$

where $a > 0$ and $E_{\pi_0(\theta|x)}$ denotes the expectation with respect to the density $\pi_0(\theta|x)$. It is known that $d(\pi(\theta|x), \pi_0(\theta|x)) \geq 0$ for all $\pi(\theta|x), \pi_0(\theta|x), a > 0$ and $d(\pi(\theta|x), \pi_0(\theta|x)) = 0$ if and only if $\pi(\theta|x) = \pi_0(\theta|x)$. Please note that the case $a = 1$ is defined by letting $a \to 1$. Other values of $a$ of a particular interest are $a = 0, 0.5, 2$ and $\infty$ (van Erven and Harremoës, 2014 [20]). For further properties of Rényi divergence consult, for example, Li and Turner (2016) [21].

Rényi divergence belongs to the following general class of family of divergence measures called the $(h, \phi)$ divergence (Menéndez et al., 1995 [17]).

**Definition 1.** *Let $h$ be a differentiable increasing real function mapping from $\left[0, \phi(0) + \lim_{t \to \infty} \frac{\phi(t)}{t}\right]$ to $[0, \infty)$. The $(h, \phi)$ divergence measure between two posterior distributions $\pi(\theta|x)$ and $\pi_0(\theta|x)$ is defined as*

$$d_\phi^h(\pi(\theta|x), \pi_0(\theta|x)) = h(d_\phi(\pi(\theta|x), \pi_0(\theta|x))),$$

*where $d_\phi(\pi(\theta|x), \pi_0(\theta|x))$ is the $\phi$ divergence defined in (3).*

Please note that Rényi divergence is a $(h, \phi)$ divergence measure with $h(x) = \frac{1}{a-1} \ln[a(a-1)x + 1]$, $\phi(x) = \frac{x^a - a(x-1) - 1}{a(a-1)}$ for $a \neq 0, 1$. To see this, from Definition 1, we have

$$h(d_\phi(\pi(\theta|x), \pi_0(\theta|x))) = \frac{1}{a-1} \ln[a(a-1)d_\phi(\pi(\theta|x), \pi_0(\theta|x)) + 1] \quad (5)$$

$$= \frac{1}{a-1} \ln\left[a(a-1) \int \pi_0(\theta|x)\left\{ \frac{\left[\frac{\pi(\theta|x)}{\pi_0(\theta|x)}\right]^a - a\left[\frac{\pi(\theta|x)}{\pi_0(\theta|x)}\right] + a - 1}{a(a-1)} \right\} d\theta + 1\right]$$

$$= \frac{1}{a-1} \ln\left[ \int \pi_0(\theta|x)\left[\frac{\pi(\theta|x)}{\pi_0(\theta|x)}\right]^a d\theta \right.$$

$$- a \int \pi_0(\theta|x)\left[\frac{\pi(\theta|x)}{\pi_0(\theta|x)}\right] d\theta$$

$$\left. + a \int \pi_0(\theta|x) d\theta - \int \pi_0(\theta|x) d\theta + 1\right]$$

$$= \frac{1}{a-1} \ln\left( E_{\pi_0(\theta|x)}\left[ \left(\frac{\pi(\theta|x)}{\pi_0(\theta|x)}\right)^a \right] \right),$$

which is Rényi divergence as defined in (4).

Similar to McCulloch (1989) [22] and Dey and Birmiwal (1994) [9] for calibrating, respectively, the Kullback-Leibler divergence and the $\phi$ divergence, it is also possible to calibrate Rényi divergence as follows. Consider a biased coin where $X = 1$ (heads) occurs with probability $p$. Then Rényi divergence between an unbiased and a biased coin is

$$d(f_0, f_1) = \frac{1}{a-1} \ln\left[2^{a-1}(p^a + (1-p)^a)\right],$$

where for $x = 0, 1$, $f_0(x) = 0.5$ and $f_1(x) = p^x(1-p)^{1-x}$. Now, setting $d(f_0, f_1) = d_0$ gives

$$2^{1-a}e^{(a-1)d_0} = p^a + (1-p)^a. \quad (6)$$

Then the number $p$ is the calibration of $d$. In general, Equation (6) needs to be solved numerically for $p$. Please note that for the case $a = 1$ (i.e., the Kullback-Leibler divergence) one may use the following explicit formula for $p$ due to McCulloch (1989) [22]:

$$p = 0.5 + 0.5\left(1 - e^{-2d_0}\right)^{1/2}. \quad (7)$$

Values of $p$ close to 1 indicate that $f_0$ and $f_1$ are quite different, while values of $p$ close to 0.5 implies that they are similar. It is restricted that $p$ is chosen so that it is between 0.5 and 1 there is a one-to-one correspondence between $p$ and $d_0$.

A motivating key fact about Rényi divergence follows from its Taylor expansion. Let

$$f(\epsilon) = d(\pi(\theta|x), \pi_0(\theta|x)) = \frac{1}{a-1} \ln\left( \int (\pi(\theta|x))^a (\pi_0(\theta|x))^{1-a} d\theta \right),$$

where $\pi(\theta|x)$ is the posterior distribution of $\theta$ given the data $x$ under the prior $\pi$ defined in (1) and (2). Assuming differentiability with respect to $\epsilon$, the Taylor expansion of $f(\epsilon)$ about $\epsilon = 0$ is given by

$$f(\epsilon) = f(0) + \epsilon \frac{\partial f(\epsilon)}{\partial \epsilon}\bigg|_{\epsilon=0} + \frac{\epsilon^2}{2} \frac{\partial^2 f(\epsilon)}{\partial \epsilon^2}\bigg|_{\epsilon=0} + \cdots.$$

Clearly, $f(0) = 0$. If integration and differentiation are interchangeable, we have

$$\frac{\partial f(\epsilon)}{\partial \epsilon} = \frac{a}{1-a} \frac{\int (\pi_0(\theta|x))^{1-a}(\pi(\theta|x))^{a-1}\frac{\partial \pi(\theta|x)}{\partial \epsilon}d\theta}{\int (\pi_0(\theta|x))^{1-a}(\pi(\theta|x))^a d\theta}.$$

Hence,

$$\frac{\partial f(\epsilon)}{\partial \epsilon}\bigg|_{\epsilon=0} = \frac{a}{1-a} \int \frac{\partial \pi(\theta|x)}{\partial \epsilon}d\theta$$

$$= \frac{a}{1-a}\frac{\partial}{\partial \epsilon}\left(\int \pi(\theta|x)d\theta\right) = \frac{a}{1-a}\frac{\partial}{\partial \epsilon}(1) = 0.$$

On the other hand,

$$\frac{\partial^2 f(\epsilon)}{\partial \epsilon^2} = \frac{\partial}{\partial \epsilon}\left(\frac{a}{1-a}\frac{\int (\pi_0(\theta|x))^{1-a}(\pi(\theta|x))^{a-1}\frac{\partial \pi(\theta|x)}{\partial \epsilon}d\theta}{\int (\pi_0(\theta|x))^{1-a}(\pi(\theta|x))^a d\theta}\right),$$

which at $\epsilon = 0$, reduces to

$$\frac{\partial^2 f(\epsilon)}{\partial \epsilon^2}\bigg|_{\epsilon=0} = -a \int \frac{\left(\frac{\partial \pi(\theta|x)}{\partial \epsilon}\right)^2}{\pi(\theta|x)}d\theta\bigg|_{\epsilon=0}$$

$$= -a \int \left(\frac{\frac{\partial \pi(\theta|x)}{\partial \epsilon}}{\pi(\theta|x)}\right)^2 \pi(\theta|x)d\theta\bigg|_{\epsilon=0}$$

$$= -aE_{\pi(\theta|x)}\left[\left(\frac{\partial \ln \pi(\theta|x)}{\partial \epsilon}\right)^2\right]\bigg|_{\epsilon=0}$$

$$= -aI_{\pi(\theta|x)}(\epsilon)\bigg|_{\epsilon=0}.$$

Here $I_{\pi(\theta|x)}(\epsilon) = E_{\pi(\theta|x)}\left[\left(\frac{\partial \ln \pi(\theta|x)}{\partial \epsilon}\right)^2\right]\bigg|_{\epsilon=0}$ is the Fisher information function for $\pi(\theta|x)$ (Lehmann and Casella, 1998 [23]). Thus, for $\epsilon \approx 0$, we have

$$d(\pi(\theta|x), \pi_0(\theta|x)) \approx -\frac{a\epsilon^2}{2}I_{\pi(\theta|x)}(\epsilon). \tag{8}$$

Please note that $\partial^2 f(\epsilon)/\partial \epsilon^2\big|_{\epsilon=0} = \partial^2 d/\partial \epsilon^2\big|_{\epsilon=0}$ is known as the local *curvature* at $\epsilon = 0$ of Rényi divergence. Formula (8) justifies the use of the curvature to measure the Bayesian robustness of the two classes of priors $\Gamma_a$ and $\Gamma_g$ as defined in (1) and (2), respectively. Also this formula provide a direct relationship between Fisher's information and the curvature of Rényi divergence.

## 3. Measuring Robustness Using Rényi Divergence

In this section, we explicitly obtain the local curvature at $\epsilon = 0$ of Rényi divergence (i.e., $\partial^2 d/\partial \epsilon^2\big|_{\epsilon=0}$), to measure the Bayesian robustness of the two classes of priors $\Gamma_a$ and

$\Gamma_g$ as defined in (1) and (2), respectively. The resulting quantities are presumably much easier to estimate than working directly with Rényi divergence.

**Theorem 1.** *For the $\epsilon$-contaminated class defined in (1), the local curvature of Rényi divergence at $\epsilon = 0$ is*

$$C_a^{\Gamma_a} = \left.\frac{\partial^2 d}{\partial \epsilon^2}\right|_{\epsilon=0} = a Var_{\pi_0(\theta|x)}\left[\frac{q(\theta)}{\pi_0(\theta)}\right],$$

*where $Var_{\pi_0(\theta|x)}$ denotes the variance with respect to $\pi_0(\theta|x)$.*

**Proof.** Under the prior $\pi$ defined in (1), the marginal $m(\theta|x)$ and the posterior distribution $\pi(\theta|x)$ can be written as

$$m(x|\pi) = (1-\epsilon)m(x|\pi_0) + \epsilon m(x|q)$$

and

$$
\begin{aligned}
\pi(\theta|x) &= \frac{f_\theta(x)\pi(\theta)}{m(x|\pi)} \\
&= \frac{f_\theta(x)((1-\epsilon)\pi_0(\theta) + \epsilon q(\theta))}{m(x|\pi)} \\
&= \lambda(x)\pi_0(\theta|x) + (1-\lambda(x))q(\theta|x),
\end{aligned}
\tag{9}
$$

where

$$\lambda(x) = (1-\epsilon)\frac{m(x|\pi_0)}{m(x|\pi)}.$$

Define

$$
\begin{aligned}
f(\epsilon) &= d(\pi(\theta|x), \pi_0(\theta|x)) \\
&= \frac{1}{a-1}\ln\left[\int (\pi(\theta|x))^a (\pi_0(\theta|x))^{1-a} d\theta\right] = \frac{1}{a-1}\ln\left[\int \gamma d\theta\right],
\end{aligned}
$$

where

$$\gamma = (\pi(\theta|x))^a (\pi_0(\theta|x))^{1-a} = (\lambda(x)\pi_0(\theta|x) + (1-\lambda(x))q(\theta|x))^a (\pi_0(\theta|x))^{1-a}.$$

Clearly,

$$\left.\gamma\right|_{\epsilon=0} = \pi_0(\theta|x) \text{ and } \left.\int \gamma\right|_{\epsilon=0} d\theta = 1.
\tag{10}$$

We have

$$\frac{\partial \gamma}{\partial \epsilon} = a\frac{m(x|q)m(x|\pi_0)(q(\theta|x) - \pi_0(\theta|x))}{[\epsilon q(\theta|x)m(x|q) + (1-\epsilon)m(x|\pi_0)\pi_0(\theta|x)][(1-\epsilon)m(x|\pi_0) + \epsilon m(x|q)]}$$

and

$$\left.\frac{\partial \gamma}{\partial \epsilon}\right|_{\epsilon=0} = a\frac{m(x|q)(q(\theta|x) - \pi_0(\theta|x))}{m(x|\pi_0)}.$$

Thus,

$$\left.\int \frac{\partial \gamma}{\partial \epsilon} d\theta\right|_{\epsilon=0} = 0.
\tag{11}$$

Now,

$$\frac{\partial^2 d}{\partial \epsilon^2} = \frac{\partial}{\partial \epsilon}\left(\frac{1}{a-1}\frac{\int \frac{\partial \gamma}{\partial \epsilon} d\theta}{\int \gamma d\theta}\right) = \frac{1}{a-1}\frac{[\int \gamma d\theta][\int \frac{\partial^2 \gamma}{\partial \epsilon^2} d\theta] - [\int \frac{\partial \gamma}{\partial \epsilon} d\theta]^2}{[\int \gamma d\theta]^2}.$$

By (10) and (11),

$$\left.\frac{\partial^2 d}{\partial \epsilon^2}\right|_{\epsilon=0} = \frac{1}{a-1} \int \left.\frac{\partial^2 \gamma}{\partial \epsilon^2}\right|_{\epsilon=0} d\theta.$$

We have

$$\left.\frac{\partial^2 \gamma}{\partial \epsilon^2}\right|_{\epsilon=0} = \left( \frac{\pi_0(\theta|x)m(x|\pi_0) - q(\theta|x)m(x|q)}{\pi_0(\theta|x)m(x|\pi_0)} + \frac{m(x|\pi_0) - m(x|q)}{m(x|\pi_0)} + \right.$$
$$\left. \frac{a\frac{m(x|q)}{m(x|\pi_0)}(q(\theta|x) - \pi_0(\theta|x))}{\pi_0(\theta|x)} \right) \times \tag{12}$$
$$a\frac{m(x|q)}{m(x|\pi_0)}(q(\theta|x) - \pi_0(\theta|x)).$$

Since

$$\frac{m(x|q)}{m(x|\pi_0)} = \frac{\int f_\theta(x)q(\theta)d\theta}{m(x|\pi_0)} = \frac{\int f_\theta(x)\pi_0(\theta)\frac{q(\theta)}{\pi_0(\theta)}d\theta}{m(x|\pi_0)}$$
$$= \int \pi_0(\theta|x)\frac{q(\theta)}{\pi_0(\theta)}d\theta$$
$$= E_{\pi_0(\theta|x)}\left[\frac{q(\theta)}{\pi_0(\theta)}\right], \tag{13}$$

from (12), we get

$$\left.\frac{\partial^2 \gamma}{\partial \epsilon^2}\right|_{\epsilon=0} = a\left( 2 - E_{\pi_0(\theta|x)}\left[\frac{q(\theta)}{\pi_0(\theta)}\right]\right) E_{\pi_0(\theta|x)}\left[\frac{q(\theta)}{\pi_0(\theta)}\right](q(\theta|x) - \pi_0(\theta|x))$$
$$- a\left(E_{\pi_0(\theta|x)}\right)^2\left[\frac{q(\theta)}{\pi_0(\theta)}\right]\left(\frac{q(\theta|x)}{\pi_0(\theta|x)}\right)(q(\theta|x) - \pi_0(\theta|x))$$
$$+ a^2\left(E_{\pi_0(\theta|x)}\right)^2\left[\frac{q(\theta)}{\pi_0(\theta)}\right]\frac{(q(\theta|x) - \pi_0(\theta|x))^2}{\pi_0(\theta|x)}.$$

Therefore,

$$\left.\frac{\partial^2 d}{\partial \epsilon^2}\right|_{\epsilon=0} = a\left( \left(E_{\pi_0(\theta|x)}\left[\frac{q(\theta)}{\pi_0(\theta)}\right]\right)^2 E_{\pi_0(\theta|x)}\left[\left(\frac{q(\theta|x)}{\pi_0(\theta|x)}\right)^2\right] \right.$$
$$\left. -\left(E_{\pi_0(\theta|x)}\left[\frac{q(\theta)}{\pi_0(\theta)}\right]\right)^2\right). \tag{14}$$

Please note that

$$\left(\frac{q(\theta|x)}{\pi_0(\theta|x)}\right)^2 = \left(\frac{q(\theta)f_\theta(x)/m(x|q)}{\pi(\theta)f_\theta(x)/m(x|\pi_0)}\right)^2 = \left(\frac{q(\theta)}{\pi(\theta)}\right)^2\left(\frac{m(x|\pi_0)}{m(x|q)}\right)^2$$

Hence, by (13),

$$E_{\pi_0(\theta|x)}\left[\left(\frac{q(\theta|x)}{\pi_0(\theta|x)}\right)^2\right] = E_{\pi_0(\theta|x)}\left[\left(\frac{q(\theta)}{\pi_0(\theta)}\right)^2\right]\frac{1}{\left(E_{\pi_0(\theta|x)}\left[\frac{q(\theta)}{\pi_0(\theta)}\right]\right)^2}. \tag{15}$$

Thus, by (14) and (15),

$$\left.\frac{\partial^2 d}{\partial \epsilon^2}\right|_{\epsilon=0} = a\left(E_{\pi_0(\theta|x)}\left[\left(\frac{q(\theta)}{\pi_0(\theta)}\right)^2\right] - \left(E_{\pi_0(\theta|x)}\left[\frac{q(\theta)}{\pi_0(\theta)}\right]\right)^2\right)$$

$$= a Var_{\pi_0(\theta|x)}\left[\frac{q(\theta)}{\pi_0(\theta)}\right].$$

□

**Theorem 2.** *For the geometric contaminated class defined in (2), the local curvature of Rényi divergence at $\epsilon = 0$ is*

$$C_a^{\Gamma_g} = \left.\frac{\partial^2 d}{\partial \epsilon^2}\right|_{\epsilon=0} = a Var_{\pi_0(\theta|x)}\left[\ln\left(\frac{q(\theta)}{\pi_0(\theta)}\right)\right],$$

$Var_{\pi_0(\theta|x)}$ *denotes the variance with respect to $\pi_0(\theta|x)$.*

**Proof.** Define

$$\gamma = (\pi(\theta|x))^a (\pi_0(\theta|x))^{1-a}.$$

Thus,

$$d = \frac{1}{a-1}\ln\left(\int \gamma d\theta\right).$$

We have

$$\frac{\partial d}{\partial \epsilon} = \frac{1}{a-1} \times \frac{\int \frac{\partial \gamma}{\partial \epsilon}d\theta}{\int \gamma d\theta}$$

and

$$\frac{\partial^2 d}{\partial \epsilon^2} = \frac{1}{a-1} \times \frac{\int \gamma d\theta \int \frac{\partial^2 \gamma}{\partial \epsilon^2}d\theta - \left(\int \frac{\partial \gamma}{\partial \epsilon}d\theta\right)^2}{\left(\int \gamma d\theta\right)^2}. \tag{16}$$

Since $\left.\gamma\right|_{\epsilon=0} = \pi_0(\theta|x)$,

$$\left.\frac{\partial^2 d}{\partial \epsilon^2}\right|_{\epsilon=0} = \left.\int \frac{\partial^2 \gamma}{\partial \epsilon^2}d\theta\right|_{\epsilon=0} - \left.\left(\int \frac{\partial \gamma}{\partial \epsilon}d\theta\right)^2\right|_{\epsilon=0}.$$

For the geometric class defined in (2),

$$\pi(\theta|x) = \frac{f_\theta(x)\pi(\theta)}{m(x|\pi)} = \frac{f_\theta(x)c(\epsilon)(\pi_0(\theta))^{1-\epsilon}(q(\theta))^\epsilon}{m(x|\pi)} \quad \text{and} \quad \pi_0(\theta|x) = \frac{f_\theta(x)\pi_0(\theta)}{m(x|\pi_0)}. \tag{17}$$

Thus,

$$\gamma = \frac{f_\theta(x)(c(\epsilon))^a(\pi_0(\theta))^{1-a\epsilon}(q(\theta))^{a\epsilon}}{(m(x|\pi))^a(m(x|\pi_0))^{1-a}}.$$

Therefore,

$$\ln(\gamma) = a\ln\left(\frac{c(\epsilon)}{m(x|\pi)}\right) - a\epsilon\ln\left(\frac{\pi_0(\theta)}{q(\theta)}\right) + \ln\frac{f_\theta(x)\pi_0(\theta)}{(m(x|\pi_0))^{1-a}}.$$

We have

$$\frac{\partial \gamma}{\partial \epsilon} = \gamma \frac{\partial \ln \gamma}{\partial \epsilon} = a\gamma \left( \frac{\partial}{\partial \epsilon} \ln \left( \frac{c(\epsilon)}{m(x|\pi)} \right) - \ln \left( \frac{\pi_0(\theta)}{q(\theta)} \right) \right). \tag{18}$$

As

$$\frac{\partial}{\partial \epsilon} \ln \left( \frac{c(\epsilon)}{m(x|\pi)} \right) = E_{\pi_0(\theta|x)} \left[ \ln \left( \frac{\pi_0(\theta)}{q(\theta)} \right) \right]$$

(Dey and Birmiwal, 1994 [9], Theorem 3.2), we get

$$\frac{\partial \gamma}{\partial \epsilon} = a\gamma \left( E_{\pi_0(\theta|x)} \left[ \ln \left( \frac{\pi_0(\theta)}{q(\theta)} \right) \right] - \ln \left( \frac{\pi_0(\theta)}{q(\theta)} \right) \right).$$

Since $\gamma \Big|_{\epsilon=0} = \pi_0(\theta|x)$, by (16) and (18), it follows that $\int \frac{\partial \gamma}{\partial \epsilon} d\theta \Big|_{\epsilon=0} = 0$ and

$$\frac{\partial^2 d}{\partial \epsilon^2} \Big|_{\epsilon=0} = \int \frac{\partial^2 \gamma}{\partial \epsilon^2} d\theta \Big|_{\epsilon=0}.$$

Now, by (18),

$$\begin{aligned}
\frac{\partial^2 \gamma}{\partial \epsilon^2} &= \frac{\partial}{\partial \epsilon} \left( a\gamma \left( E_{\pi_0(\theta|x)} \left[ \ln \left( \frac{\pi_0(\theta)}{q(\theta)} \right) \right] - \ln \left( \frac{\pi_0(\theta)}{q(\theta)} \right) \right) \right) \\
&= a\gamma \left( E_{\pi_0(\theta|x)} \left[ \ln \left( \frac{\pi_0(\theta)}{q(\theta)} \right) \right] - \ln \left( \frac{\pi_0(\theta)}{q(\theta)} \right) \right)^2.
\end{aligned}$$

Using the $\gamma \Big|_{\epsilon=0} = \pi_0(\theta|x)$ one more time, we obtain

$$\frac{\partial^2 d}{\partial \epsilon^2} \Big|_{\epsilon=0} = \int \frac{\partial^2 \gamma}{\partial \epsilon^2} \Big|_{\epsilon=0} d\theta = aVar_{\pi_0(\theta|x)} \left[ \ln \left( \frac{q(\theta)}{\pi_0(\theta)} \right) \right].$$

$\square$

The curvature of the family $(h, \phi)$ of divergence measures under classes (1) and (2) is derived in the next theorem.

**Theorem 3.** *The local curvature for the $(h, \phi)$ divergence under classes (1) and (2) are respectively given by*

*i.* $\quad C_a^{\Gamma_a} = \frac{\partial^2 d_\phi^h(\pi(\theta|x), \pi_0(\theta|x))}{\partial \epsilon^2} \Big|_{\epsilon=0} = a\phi''(1) Var_{\pi_0(\theta|x)} \left[ \frac{q(\theta)}{\pi_0(\theta)} \right],$

*ii.* $\quad C_a^{\Gamma_g} = \frac{\partial^2 d_\phi^h(\pi(\theta|x), \pi_0(\theta|x))}{\partial \epsilon^2} \Big|_{\epsilon=0} = a\phi''(1) Var_{\pi_0(\theta|x)} \left[ \ln \left( \frac{q(\theta)}{\pi_0(\theta)} \right) \right],$

*where $\phi''(1)$ is the second derivation of smooth convex function $\phi$ at 1.*

**Proof.** To prove (i), from Equation (5), we have

$$\begin{aligned}
\frac{\partial d_\phi^h(\pi(\theta|x), \pi_0(\theta|x))}{\partial \epsilon} &= \frac{1}{a-1} \left[ \frac{a(a-1)\frac{\partial d_\phi(\pi(\theta|x), \pi_0(\theta|x))}{\partial \epsilon}}{a(a-1)d_\phi(\pi(\theta|x), \pi_0(\theta|x)) + 1} \right] \\
&= a\frac{\frac{\partial d_\phi(\pi(\theta|x), \pi_0(\theta|x))}{\partial \epsilon}}{a(a-1)d_\phi(\pi(\theta|x), \pi_0(\theta|x)) + 1}.
\end{aligned}$$

Now, we get

$$\frac{\partial^2 d_\phi^h(\pi(\theta|x), \pi_0(\theta|x))}{\partial \epsilon^2} = a \left\{ \frac{\left[a(a-1)d_\phi(\pi(\theta|x), \pi_0(\theta|x)) + 1\right] \frac{\partial^2 d_\phi(\pi(\theta|x), \pi_0(\theta|x))}{\partial \epsilon^2}}{\left[a(a-1)d_\phi(\pi(\theta|x), \pi_0(\theta|x)) + 1\right]^2} \right.$$
$$\left. - \frac{a(a-1)\left[\frac{\partial d_\phi^h(\pi(\theta|x), \pi_0(\theta|x))}{\partial \epsilon}\right]^2}{\left[a(a-1)d_\phi(\pi(\theta|x), \pi_0(\theta|x)) + 1\right]^2} \right\}. \tag{19}$$

From Dey and Birmiwal (1994, Thm 3.1) [9], under class (1), we have

$$\left. \frac{\partial d_\phi(\pi(\theta|x), \pi_0(\theta|x))}{\partial \epsilon} \right|_{\epsilon=0} = 0$$

and

$$\left. \frac{\partial^2 d_\phi(\pi(\theta|x), \pi_0(\theta|x))}{\partial \epsilon^2} \right|_{\epsilon=0} = \phi''(1) Var_{\pi_0(\theta|x)} \left[\frac{q(\theta)}{\pi_0(\theta)}\right].$$

Therefore,

$$\left. \frac{\partial^2 d_\phi^h(\pi(\theta|x), \pi_0(\theta|x))}{\partial \epsilon^2} \right|_{\epsilon=0} = \frac{[0+1]a\phi''(1)Var_{\pi_0(\theta|x)}\left[\frac{q(\theta)}{\pi_0(\theta)}\right] - 0}{1},$$

and the proof of (i) is concluded. To prove (ii), from Dey and Birmiwal (1994, Thm 3.2.) [9], under class (2), we have

$$\left. \frac{\partial d_\phi(\pi(\theta|x), \pi_0(\theta|x))}{\partial \epsilon} \right|_{\epsilon=0} = 0$$

and

$$\left. \frac{\partial^2 d_\phi(\pi(\theta|x), \pi_0(\theta|x))}{\partial \epsilon^2} \right|_{\epsilon=0} = \phi''(1) Var_{\pi_0(\theta|x)} \left[\ln\left(\frac{q(\theta)}{\pi_0(\theta)}\right)\right].$$

Similar to the proof of (i), by considering the above equations in (19) the proof of (ii) is concluded. □

Please note that since for Rényi divergence $\phi(x) = \frac{x^a - a(x-1) - 1}{a(a-1)}$, we have $\phi''(1) = 1$. This implies that Theorems 1 and 2 can be obtained by Theorem 3. However, the proofs of Theorems 1 and 2 are more general and could be applied to cases that are not a member of $(h, \phi)$ divergence.

## 4. Examples

In this section, the derived results are explained through three examples: the Bernoulli model, the multinomial model and the location normal model. In each example, the curvature values for the two classes (1) and (2) are reported. Additionally, in Example 1, we computed Rényi divergence between $\pi(\theta|x)$ and $\pi_0(\theta|x)$ and reported the calibrated value $p$ as described in (6) and (7). Recall that curvature values close to zero indicate robustness of the used prior whereas larger values suggest lack of robustness. On the other hand, values of $p$ close to 0.5 suggest robustness whereas values of $p$ close to 1 means absence of robustness.

**Example 1** (Bernoulli Model). *Suppose* $\mathbf{x} = (x_1, \ldots, x_n)$ *is a sample from a Bernoulli distribution with a parameter $\theta$. Let the prior $\pi_0(\theta)$ be Beta($\alpha, \beta$), i.e.,*

$$\pi(\theta) = \frac{\Gamma(\alpha + \beta)}{\Gamma(\alpha)\Gamma(\beta)} \theta^{\alpha-1}(1-\theta)^{\beta-1}.$$

*Thus, $\pi_0(\theta|\mathbf{x})$ is*

$$Beta(\alpha + t, \beta + n - t),$$

*where $t = \sum_{i=1}^{n} x_i$. Let $q(\theta)$ be $Beta(c\alpha, c\beta)$ for $c > 0$.*

Now consider the two samples $\mathbf{x} = (0,0,1,1,0,1,1,1,1,0,0,0,1,0,1,0,1,1,0,1)$ and $\mathbf{x} = (0,0,1,1,0,1,1,1,1,0,0,0,1,0,1,0,1,1,0,1,1,0,1,0,0,0,0,0,1,0,0,1,0,0,1,1,1,0,1,0,1,$ $1,1,1,1,1,0,0,1,1)$ of sizes $n = 20$ and $n = 50$ generated from *Bernoulli(0.5)*. For comparison purposes, we consider several values of $\alpha$, $\beta$ and $c$. Although it is possible to find exact formulas of the curvature by some algebraic manipulation, it looks more convenient to use a Monte Carlo approach in this example. The computational steps are summarized in Algorithm 1.

---

**Algorithm 1** Computing curvature based on Monte Carlo approach

---

1. For $s = 1, \cdots, 10^6$, generate $\theta^{(s)}$ from the posterior $\pi_0(\theta|\mathbf{x})$.
2. For each $\theta^{(s)}$, find $q(\theta^{(s)})$ and $\pi_0(\theta^{(s)})$.
3. Compute the sample variance of the $10^6$ values of $q(\theta^{(s)})/\pi_0(\theta^{(s)})$. Denote this value by $\hat{Var}_{\pi_0(\theta|\mathbf{x})}(q(\theta)/\pi_0(\theta))$.
4. Return $a\hat{Var}_{\pi_0(\theta|\mathbf{x})}(q(\theta)/\pi_0(\theta))$ as the curvature value under class (1).
5. Compute the sample variance of the $10^6$ values of $\ln\left(q(\theta^{(s)})/\pi_0(\theta^{(s)})\right)$. Denote this values by $\hat{Var}_{\pi_0(\theta|\mathbf{x})}(\ln(q(\theta)/\pi_0(\theta)))$.
6. Return $a\hat{Var}_{\pi_0(\theta|\mathbf{x})}(\ln(q(\theta)/\pi_0(\theta)))$ as the curvature value under class (2).

---

The values of the curvature for both classes (1) and (2) are reported in Table 1. Remarkably, for the cases when $\alpha = \beta = 1$ (uniform prior on $[0,1]$) and $\alpha = \beta = 0.5$ (Jeffreys' prior), the curvature values are prominently small for all values of $c$. Also, it is clear that when $c = 1$, the curvature values are 0. It worth noticing here that when fixing the parameters $\alpha$, $\beta$ and $c$, the curvature decrease by increasing the sample size. This supports the fact that the effect of the prior dissipates with increasing the sample.

While it is easier to quantify the curvature based on Theorems 1 and 2, in this example, for comparison purposes, we computed Rényi divergence between $\pi(\theta|\mathbf{x})$ and $\pi_0(\theta|\mathbf{x})$ under classes (1) and (2). It can be shown that under class (1) in (9), $\pi(\theta|\mathbf{x}) = \lambda(\mathbf{x})Beta(\alpha + t, \beta + n - t) + (1 - \lambda(\mathbf{x}))Beta(c\alpha + t, c\beta + n - t)$, where

$$\lambda(\mathbf{x}) = (1 - \epsilon)\frac{\frac{\Gamma(\alpha+\beta)}{\Gamma(\alpha)\Gamma(\beta)}\frac{\Gamma(\alpha+t)\Gamma(\beta-t+n)}{\Gamma(\alpha+\beta+n)}}{(1-\epsilon)\frac{\Gamma(\alpha+\beta)}{\Gamma(\alpha)\Gamma(\beta)}\frac{\Gamma(\alpha+t)\Gamma(\beta-t+n)}{\Gamma(\alpha+\beta+n)} + \epsilon\frac{\Gamma(c\alpha+c\beta)}{\Gamma(c\alpha)\Gamma(c\beta)}\frac{\Gamma(c\alpha+t)\Gamma(c\beta-t+n)}{\Gamma(c\alpha+c\beta+n)}}.$$

Also, from (17), it can be easily concluded that the posterior $\pi(\theta|\mathbf{x})$ under class (2) is obtained as

$$\pi(\theta|\mathbf{x}) = K \times \frac{\theta^t(1-\theta)^{n-t}[Beta(\alpha,\beta)]^{1-\epsilon}[Beta(c\alpha,c\beta)]^{\epsilon}}{\left[\frac{\Gamma(\alpha+\beta)}{\Gamma(\alpha)\Gamma(\beta)}\right]^{(1-\epsilon)}\left[\frac{\Gamma(c\alpha+c\beta)}{\Gamma(c\alpha)\Gamma(c\beta)}\right]^{\epsilon}},$$

$$K = \frac{\Gamma(t + (1-\epsilon)(\alpha-1) + \epsilon(c\alpha-1) + 1)\Gamma(n - t + (1-\epsilon)(\beta-1) + \epsilon(c\beta-1) + 1)}{\Gamma((1-\epsilon)(\alpha+\beta-2) + \epsilon(c\alpha+c\beta-2) + n + 2)}.$$

**Table 1.** Values of the local curvature for two classes $\Gamma_a$ and $\Gamma_g$ for a sample generated from Bernoulli(0.5).

| $n$ | $\begin{pmatrix}\alpha\\\beta\end{pmatrix}$ | $c$ | $a = 0.5$ | | $a = 1$ | | $a = 2$ | |
|---|---|---|---|---|---|---|---|---|
| | | | $C_a^{\Gamma_a}$ | $C_a^{\Gamma_g}$ | $C_a^{\Gamma_a}$ | $C_a^{\Gamma_g}$ | $C_a^{\Gamma_a}$ | $C_a^{\Gamma_g}$ |
| 20 | $\begin{pmatrix}0.5\\0.5\end{pmatrix}$ | 0.5 | $8 \times 10^{-5}$ | 0.0002 | 0.0001 | 0.0004 | 0.0003 | 0.0008 |
| | | 1 | 0 | 0 | 0 | 0 | 0 | 0 |
| | | 1.5 | 0.0003 | 0.0002 | 0.0006 | 0.0004 | 0.0013 | 0.0008 |
| | | 3 | 0.0098 | 0.0033 | 0.0196 | 0.0067 | 0.0393 | 0.0135 |
| | | 5 | 0.0531 | 0.0135 | 0.1062 | 0.0271 | 0.2125 | 0.0543 |
| | $\begin{pmatrix}1\\1\end{pmatrix}$ | 0.5 | 0.0003 | 0.0007 | 0.0007 | 0.0015 | 0.0014 | 0.0030 |
| | | 1 | 0 | 0 | 0 | 0 | 0 | 0 |
| | | 1.5 | 0.0010 | 0.0007 | 0.0021 | 0.0015 | 0.0042 | 0.0030 |
| | | 3 | 0.0241 | 0.0121 | 0.0483 | 0.0243 | 0.0967 | 0.0486 |
| | | 5 | 0.1065 | 0.0486 | 0.2130 | 0.0972 | 0.4260 | 0.1945 |
| | $\begin{pmatrix}1\\3\end{pmatrix}$ | 0.5 | 0.0265 | 0.0235 | 0.0530 | 0.0470 | 0.1060 | 0.0941 |
| | | 1 | 0 | 0 | 0 | 0 | 0 | 0 |
| | | 1.5 | 0.0171 | 0.0235 | 0.0342 | 0.0470 | 0.0684 | 0.0941 |
| | | 3 | 0.1061 | 0.3767 | 0.2122 | 0.7535 | 0.4244 | 1.5070 |
| | | 5 | 0.1660 | 1.5070 | 0.3320 | 3.0141 | 0.6641 | 6.0282 |
| | $\begin{pmatrix}3\\1\end{pmatrix}$ | 0.5 | 0.0089 | 0.0113 | 0.0179 | 0.0227 | 0.0133 | 0.0454 |
| | | 1 | 0 | 0 | 0 | 0 | 0 | 0 |
| | | 1.5 | 0.0108 | 0.0113 | 0.0216 | 0.0227 | 0.0433 | 0.0454 |
| | | 3 | 0.1162 | 0.1819 | 0.2324 | 0.3638 | 0.4648 | 0.7277 |
| | | 5 | 0.2774 | 0.7277 | 0.5548 | 1.4555 | 1.1096 | 2.9110 |
| 50 | $\begin{pmatrix}0.5\\0.5\end{pmatrix}$ | 0.5 | $10^{-5}$ | $4 \times 10^{-5}$ | $3 \times 10^{-5}$ | $8 \times 10^{-5}$ | $6 \times 10^{-5}$ | 0.0001 |
| | | 1 | 0 | 0 | 0 | 0 | 0 | 0 |
| | | 1.5 | $6 \times 10^{-5}$ | $4 \times 10^{-5}$ | 0.0001 | $8 \times 10^{-5}$ | 0.0002 | 0.0001 |
| | | 3 | 0.0022 | 0.0006 | 0.0044 | 0.0013 | 0.0089 | 0.0026 |
| | | 5 | 0.0139 | 0.0026 | 0.0279 | 0.0052 | 0.0559 | 0.0104 |
| | $\begin{pmatrix}1\\1\end{pmatrix}$ | 0.5 | $6 \times 10^{-5}$ | 0.0001 | 0.0001 | 0.0003 | 0.0002 | 0.0006 |
| | | 1 | 0 | 0 | 0 | 0 | 0 | 0 |
| | | 1.5 | 0.0002 | 0.0001 | 0.0004 | 0.0003 | 0.0009 | 0.0006 |
| | | 3 | 0.0066 | 0.0024 | 0.0132 | 0.0049 | 0.0265 | 0.0099 |
| | | 5 | 0.0359 | 0.0099 | 0.0718 | 0.0198 | 0.1437 | 0.0397 |
| | $\begin{pmatrix}1\\3\end{pmatrix}$ | 0.5 | 0.0106 | 0.0112 | 0.0212 | 0.0225 | 0.0425 | 0.0451 |
| | | 1 | 0 | 0 | 0 | 0 | 0 | 0 |
| | | 1.5 | 0.0087 | 0.0112 | 0.0174 | 0.0225 | 0.0349 | 0.0451 |
| | | 3 | 0.0490 | 0.1805 | 0.0980 | 0.3610 | 0.1960 | 0.7221 |
| | | 5 | 0.0535 | 0.7221 | 0.1070 | 1.4442 | 0.2140 | 2.8885 |
| | $\begin{pmatrix}3\\1\end{pmatrix}$ | 0.5 | 0.0042 | 0.0060 | 0.0084 | 0.0121 | 0.0169 | 0.0243 |
| | | 1 | 0 | 0 | 0 | 0 | 0 | 0 |
| | | 1.5 | 0.0061 | 0.0060 | 0.0123 | 0.0121 | 0.0247 | 0.0243 |
| | | 3 | 0.0672 | 0.0972 | 0.1344 | 0.1944 | 0.2688 | 0.3889 |
| | | 5 | 0.1407 | 0.3889 | 0.2814 | 0.7779 | 0.5628 | 1.5559 |

Please note that since $d(\pi(\theta|\mathbf{x}), \pi_0(\theta|\mathbf{x})) = \frac{1}{a-1} \ln\left( E_{\pi_0(\theta|\mathbf{x})}\left[ \left( \frac{\pi(\theta|\mathbf{x})}{\pi_0(\theta|\mathbf{x})} \right)^a \right] \right)$, it possible to compute the distance based on a Monte Carlo approach. When $a = 1$, $d(\pi(\theta|\mathbf{x}), \pi_0(\theta|\mathbf{x})) = E_{\pi_0(\theta|\mathbf{x})}\left[ \frac{\pi(\theta|\mathbf{x})}{\pi_0(\theta|\mathbf{x})} \ln\left( \frac{\pi(\theta|\mathbf{x})}{\pi_0(\theta|\mathbf{x})} \right) \right]$, the Kullback-Leibler divergence. We also calibrated Rényi divergence values as described in (6) and (7).To save space, the results based on class (1) and (2) of the sample of size $n = 20$ are reported in Tables 2 and 3, respectively.

**Table 2.** Values of $d_0$ and $p$ in (6) (for $a \neq 1$) and (7) (for $a = 1$) under class (1) for a sample generated from Bernoulli(0.5).

| $\binom{\alpha}{\beta}$ | $c$ | | $a = 0.5$ | | | $a = 1$ | | | $a = 2$ | | |
| --- | --- | --- | --- | --- | --- | --- | --- | --- | --- | --- | --- |
| | | | $\epsilon = 0.05$ | $\epsilon = 0.5$ | $\epsilon = 1$ | $\epsilon = 0.05$ | $\epsilon = 0.5$ | $\epsilon = 1$ | $\epsilon = 0.05$ | $\epsilon = 0.5$ | $\epsilon = 1$ |
| $\binom{0.5}{0.5}$ | 0.5 | $d_0$ | $2 \times 10^{-7}$ | $4 \times 10^{-6}$ | $9 \times 10^{-5}$ | $5 \times 10^{-7}$ | $3 \times 10^{-5}$ | 0.0002 | $10^{-6}$ | $7 \times 10^{-5}$ | 0.0004 |
| | | $p$ | (0.5003) | (0.5022) | (0.51) | (0.5005) | (0.5042) | (0.5107) | (0.5003) | (0.5041) | (0.5106) |
| | 1 | $d_0$ | 0 | 0 | 0 | 0 | 0 | 0 | 0 | 0 | 0 |
| | | $p$ | (0.5) | (0.5) | (0.5) | (0.5) | (0.5) | (0.5) | (0.5) | (0.5) | (0.5) |
| | 1.5 | $d_0$ | $2 \times 10^{-6}$ | $4 \times 10^{-5}$ | 0.0001 | $2 \times 10^{-7}$ | $5 \times 10^{-5}$ | 0.0001 | $3 \times 10^{-7}$ | 0.0001 | 0.0003 |
| | | $p$ | (0.5013) | (0.5068) | (0.5104) | (0.5003) | (0.5054) | (0.5098) | (0.5003) | (0.5053) | (0.5096) |
| | 3 | $d_0$ | $4 \times 10^{-6}$ | 0.0004 | 0.0015 | $10^{-5}$ | 0.0012 | 0.0028 | $3 \times 10^{-5}$ | 0.0023 | 0.0054 |
| | | $p$ | (0.5022) | (0.5204) | (0.5393) | (0.5031) | (0.5244) | (0.5379) | (0.5030) | (0.5239) | (0.5367) |
| | 5 | $d_0$ | $5 \times 10^{-5}$ | 0.0019 | 0.0055 | 0.0001 | 0.0048 | 0.0102 | 0.0002 | 0.0090 | 0.0181 |
| | | $p$ | (0.5071) | (0.5437) | (0.5741) | (0.5074) | (0.5493) | (0.5711) | (0.5074) | (0.5476) | (0.5676) |
| $\binom{1}{1}$ | 0.5 | $d_0$ | $7 \times 10^{-7}$ | $5 \times 10^{-5}$ | 0.0003 | $10^{-6}$ | 0.0001 | 0.0008 | $3 \times 10^{-6}$ | 0.0002 | 0.0017 |
| | | $p$ | (0.5007) | (0.5071) | (0.5193) | (0.5009) | (0.5083) | (0.5204) | (0.5007) | (0.5084) | (0.5207) |
| | 1 | $d_0$ | 0 | 0 | 0 | 0 | 0 | 0 | 0 | 0 | 0 |
| | | $p$ | (0.5) | (0.5) | (0.5) | (0.5) | (0.5) | (0.5) | (0.5) | (0.5) | (0.5) |
| | 1.5 | $d_0$ | $2 \times 10^{-7}$ | $7 \times 10^{-5}$ | 0.0003 | $10^{-6}$ | 0.0002 | 0.0006 | $2 \times 10^{-6}$ | 0.0003 | 0.0013 |
| | | $p$ | (0.5003) | (0.5084) | (0.5193) | (0.5008) | (0.5100) | (0.5185) | (0.5007) | (0.51) | (0.5180) |
| | 3 | $d_0$ | $10^{-5}$ | 0.0013 | 0.0050 | $5 \times 10^{-5}$ | 0.0034 | 0.0092 | 0.0001 | 0.0065 | 0.0165 |
| | | $p$ | (0.5042) | (0.5364) | (0.5706) | (0.5050) | (0.5416) | (0.5677) | (0.505) | (0.5405) | (0.5645) |
| | 5 | $d_0$ | $8 \times 10^{-5}$ | 0.0050 | 0.0167 | 0.0002 | 0.0124 | 0.0297 | 0.0004 | 0.0225 | 0.0494 |
| | | $p$ | (0.5092) | (0.5708) | (0.6279) | (0.5107) | (0.5785) | (0.6201) | (0.5106) | (0.5755) | (0.6125) |
| $\binom{1}{3}$ | 0.5 | $d_0$ | $2 \times 10^{-5}$ | 0.0032 | 0.0133 | $7 \times 10^{-5}$ | 0.0067 | 0.0282 | 0.0001 | 0.0145 | 0.0623 |
| | | $p$ | (0.5053) | (0.5565) | (0.6143) | (0.5059) | (0.5580) | (0.6171) | (0.5060) | (0.5604) | (0.6268) |
| | 1 | $d_0$ | 0 | 0 | 0 | 0 | 0 | 0 | 0 | 0 | 0 |
| | | $p$ | (0.5) | (0.5) | (0.5) | (0.5) | (0.5) | (0.5) | (0.5) | (0.5) | (0.5) |
| | 1.5 | $d_0$ | $2 \times 10^{-5}$ | 0.0023 | 0.0104 | $3 \times 10^{-5}$ | 0.0045 | 0.0199 | $7 \times 10^{-5}$ | 0.0088 | 0.0370 |
| | | $p$ | (0.505) | (0.5484) | (0.6015) | (0.5044) | (0.5476) | (0.5989) | (0.5044) | (0.5472) | (0.5971) |
| | | $p$ | (0.5081) | (0.5846) | (0.6878) | (0.5077) | (0.5834) | (0.6795) | (0.5077) | (0.5833) | (0.6793) |
| | 3 | $d_0$ | 0.0001 | 0.0175 | 01213 | 0.0002 | 0.0349 | 0.2125 | 0.0005 | 0.0691 | 0.3421 |
| | | $p$ | (0.5119) | (0.6308) | (0.8181) | (0.5115) | (0.6299) | (0.7942) | (0.5117) | (0.6337) | (0.8193) |
| | 5 | $d_0$ | 0.0002 | 0.0308 | 0.3423 | 0.0004 | 0.0638 | 0.5519 | 0.0008 | 0.1337 | 0.6003 |
| | | $p$ | (0.5145) | (0.6715) | (0.9536) | (0.5146) | (0.6731) | (0.9087) | (0.5144) | (0.6891) | (0.9535) |
| $\binom{3}{1}$ | 0.5 | $d_0$ | $7 \times 10^{-6}$ | 0.0012 | 0.0063 | $2 \times 10^{-5}$ | 0.0027 | 0.0135 | $5 \times 10^{-5}$ | 0.0057 | 0.0295 |
| | | $p$ | (0.5026) | (0.5356) | (0.5791) | (0.5036) | (0.5369) | (0.5816) | (0.5034) | (0.5379) | (0.5866) |
| | 1 | $d_0$ | 0 | 0 | 0 | 0 | 0 | 0 | 0 | 0 | 0 |
| | | $p$ | (0.5) | (0.5) | (0.5) | (0.5) | (0.5) | (0.5) | (0.5) | (0.5) | (0.5) |
| | 1.5 | $d_0$ | $10^{-5}$ | 0.0013 | 0.0051 | $2 \times 10^{-5}$ | 0.0025 | 0.0096 | $4 \times 10^{-5}$ | 0.0048 | 0.0180 |
| | | $p$ | (0.5040) | (0.5364) | (0.5713) | (0.5034) | (0.5354) | (0.5692) | (0.5032) | (0.535) | (0.5674) |
| | 3 | $d_0$ | 0.0001 | 0.0139 | 0.0600 | 0.0002 | 0.0286 | 0.1054 | 0.0005 | 0.0505 | 0.1711 |
| | | $p$ | (0.5125) | (0.6168) | (0.7342) | (0.5117) | (0.6143) | (0.7180) | (0.5119) | (0.6137) | (0.7160) |
| | 5 | $d_0$ | 0.0003 | 0.0340 | 0.1724 | 0.0006 | 0.0657 | 0.2786 | 0.0012 | 0.1231 | 0.4062 |
| | | $p$ | (0.5196) | (0.68) | (0.865) | (0.5183) | (0.6754) | (0.8268) | (0.5177) | (0.6809) | (0.8539) |

**Table 3.** Values of $d_0$ and $p$ in (6) (for $a \neq 1$) and (7) (for $a = 1$) under class (2) for a sample generated from Bernoulli(0.5).

| $\binom{\alpha}{\beta}$ | $c$ | | a = 0.5 | | | a = 1 | | | a = 2 | | |
|---|---|---|---|---|---|---|---|---|---|---|---|
| | | | $\epsilon = 0.05$ | $\epsilon = 0.5$ | $\epsilon = 1$ | $\epsilon = 0.05$ | $\epsilon = 0.5$ | $\epsilon = 1$ | $\epsilon = 0.05$ | $\epsilon = 0.5$ | $\epsilon = 1$ |
| $\binom{0.5}{0.5}$ | 0.5 | $d_0$ | $2 \times 10^{-7}$ | $2 \times 10^{-5}$ | $9 \times 10^{-5}$ | $10^{-6}$ | $5 \times 10^{-5}$ | 0.0002 | $2 \times 10^{-6}$ | 0.0001 | 0.0004 |
| | | $p$ | (0.5003) | (0.5043) | (0.51) | (0.5007) | (0.5054) | (0.5107) | (0.5007) | (0.5053) | (0.5106) |
| | 1 | $d_0$ | 0 | 0 | 0 | 0 | 0 | 0 | 0 | 0 | 0 |
| | | $p$ | (0.5) | (0.5) | (0.5) | (0.5) | (0.5) | (0.5) | (0.5) | (0.5) | (0.5) |
| | 1.5 | $d_0$ | $7 \times 10^{-7}$ | $3 \times 10^{-5}$ | 0.0001 | $3 \times 10^{-8}$ | $4 \times 10^{-5}$ | 0.0001 | $6 \times 10^{-8}$ | $9 \times 10^{-5}$ | 0.0003 |
| | | $p$ | (0.5007) | (0.5053) | (0.5104) | (0.5001) | (0.5048) | (0.5098) | (0.5) | (0.505) | (0.5096) |
| | 3 | $d_0$ | $6 \times 10^{-6}$ | 0.0004 | 0.0015 | $6 \times 10^{-6}$ | 0.0007 | 0.0028 | $10^{-5}$ | 0.0014 | 0.0054 |
| | | $p$ | (0.5023) | (0.5204) | (0.5393) | (0.5017) | (0.5195) | (0.5379) | (0.5014) | (0.5191) | (0.5367) |
| | 5 | $d_0$ | $2 \times 10^{-5}$ | 0.0015 | 0.0055 | $2 \times 10^{-5}$ | 0.0028 | 0.0102 | $5 \times 10^{-5}$ | 0.0054 | 0.0181 |
| | | $p$ | (0.5045) | (0.5393) | (0.5741) | (0.5038) | (0.5379) | (0.5711) | (0.5036) | (0.5367) | (0.5676) |
| $\binom{1}{1}$ | 0.5 | $d_0$ | $6 \times 10^{-8}$ | $8 \times 10^{-5}$ | 0.0003 | $2 \times 10^{-6}$ | 0.0002 | 0.0008 | $5 \times 10^{-6}$ | 0.0004 | 0.0017 |
| | | $p$ | (0.5) | (0.5095) | (0.5193) | (0.5012) | (0.5101) | (0.5204) | (0.5011) | (0.5103) | (0.5207) |
| | 1 | $d_0$ | 0 | 0 | 0 | 0 | 0 | 0 | 0 | 0 | 0 |
| | | $p$ | (0.5) | (0.5) | (0.5) | (0.5) | (0.5) | (0.5) | (0.5) | (0.5) | (0.5) |
| | 1.5 | $d_0$ | $10^{-6}$ | 0.0001 | 0.0003 | $8 \times 10^{-7}$ | 0.0001 | 0.0006 | $10^{-6}$ | 0.0003 | 0.0013 |
| | | $p$ | (0.5013) | (0.51) | (0.5193) | (0.5006) | (0.5093) | (0.5185) | (0.5007) | (0.5093) | (0.5180) |
| | 3 | $d_0$ | $10^{-5}$ | 0.0014 | 0.0050 | $2 \times 10^{-5}$ | 0.0026 | 0.0092 | $5 \times 10^{-5}$ | 0.0048 | 0.0165 |
| | | $p$ | (0.5043) | (0.5373) | (0.5706) | (0.5035) | (0.5360) | (0.5677) | (0.5037) | (0.535) | (0.5645) |
| | 5 | $d_0$ | $6 \times 10^{-5}$ | 0.0050 | 0.0167 | 0.0001 | 0.0092 | 0.0297 | 0.0002 | 0.0165 | 0.0494 |
| | | $p$ | (0.5081) | (0.5706) | (0.6279) | (0.5074) | (0.5677) | (0.6201) | (0.5073) | (0.5645) | (0.6125) |
| $\binom{1}{3}$ | 0.5 | $d_0$ | $2 \times 10^{-5}$ | 0.0030 | 0.0133 | $6 \times 10^{-5}$ | 0.0064 | 0.0282 | 0.0001 | 0.0135 | 0.0623 |
| | | $p$ | (0.505) | (0.5555) | (0.6143) | (0.5056) | (0.5566) | (0.6171) | (0.5054) | (0.5583) | (0.6268) |
| | 1 | $d_0$ | 0 | 0 | 0 | 0 | 0 | 0 | 0 | 0 | 0 |
| | | $p$ | (0.5) | (0.5) | (0.5) | (0.5) | (0.5) | (0.5) | (0.5) | (0.5) | (0.5) |
| | 1.5 | $d_0$ | $3 \times 10^{-5}$ | 0.0028 | 0.0104 | $5 \times 10^{-5}$ | 0.0053 | 0.0199 | 0.0001 | 0.0103 | 0.0370 |
| | | $p$ | (0.5059) | (0.5527) | (0.6015) | (0.5022) | (0.5517) | (0.5989) | (0.5053) | (0.5509) | (0.5971) |
| | 3 | $d_0$ | 0.0004 | 0.0373 | 0.1213 | 0.0008 | 0.0690 | 0.2125 | 0.0017 | 0.1210 | 0.3421 |
| | | $p$ | (0.5216) | (0.6878) | (0.8181) | (0.5211) | (0.6795) | (0.7942) | (0.5209) | (0.6793) | (0.8193) |
| | 5 | $d_0$ | 0.0018 | 0.1213 | 0.3423 | 0.0034 | 0.2125 | 0.5519 | 0.0067 | 0.3421 | 0.6003 |
| | | $p$ | (0.5425) | (0.8181) | (0.9536) | (0.5417) | (0.7942) | (0.9087) | (0.5411) | (0.8193) | (0.9535) |
| $\binom{3}{1}$ | 0.5 | $d_0$ | $10^{-5}$ | 0.0014 | 0.0063 | $3 \times 10^{-5}$ | 0.0031 | 0.0135 | $6 \times 10^{-5}$ | 0.0065 | 0.0295 |
| | | $p$ | (0.5031) | (0.5381) | (0.5791) | (0.5040) | (0.5394) | (0.5816) | (0.5039) | (0.5403) | (0.5866) |
| | 1 | $d_0$ | 0 | 0 | 0 | 0 | 0 | 0 | 0 | 0 | 0 |
| | | $p$ | (0.5) | (0.5) | (0.5) | (0.5) | (0.5) | (0.5) | (0.5) | (0.5) | (0.5) |
| | 1.5 | $d_0$ | $10^{-5}$ | 0.0014 | 0.0052 | $2 \times 10^{-5}$ | 0.0025 | 0.0096 | $4 \times 10^{-5}$ | 0.0049 | 0.0180 |
| | | $p$ | (0.5041) | (0.5376) | (0.5720) | (0.5034) | (0.5359) | (0.5692) | (0.5033) | (0.5353) | (0.5674) |
| | 3 | $d_0$ | 0.0002 | 0.0185 | 0.0604 | 0.0004 | 0.0338 | 0.1054 | 0.0008 | 0.0596 | 0.1711 |
| | | $p$ | (0.5153) | (0.6341) | (0.735) | (0.5145) | (0.6278) | (0.7180) | (0.5143) | (0.6239) | (0.7160) |
| | 5 | $d_0$ | 0.0008 | 0.0604 | 0.1724 | 0.0016 | 0.1054 | 0.2786 | 0.0032 | 0.1711 | 0.4074 |
| | | $p$ | (0.53) | (0.735) | (0.865) | (0.5289) | (0.7180) | (0.8268) | (0.5284) | (0.7160) | (0.8545) |

Please note that from (8), by multiplying the curvature value in Table 1 by $\epsilon^2/2$, one may get the value of the corresponding distance in Tables 2 and 3. For instance, setting $\alpha = 1, \beta = 3, c = 0.5, a = 0.5$ in Table 1, gives $C_a^{\Gamma_a} = 0.0265$. The corresponding distance is $0.0265 \times 0.5^2/2 = 0.0033$, which close to the one reported in Table 2.

Now we consider the Australian AIDS survival data, available in the **R** package "Mass". There are 2843 patients diagnosed with AIDS in Australia before 1 July 1991. The

data frame contains the following columns: state, sex, date of diagnosis, date of death at end of observation, status ("$A = 0$" (alive) or "$D = 1$" (dead) at end of observation), reported transmission category, and age at diagnosis. There are 1082 and 1761 alive and dead cases. We consider the values of column status. Under the prior distribution given above, the values of the curvatures for two classes (1) and (2) are summarized in Table 4 for a random sample of size $n = 20$ and for the whole data. The sampled data is $\mathbf{x} = (1, 1, 1, 0, 1, 0, 0, 0, 1, 1, 0, 0, 1, 0, 1, 0, 0, 1, 1, 0)$. It interesting to notice that unlike the sample of size $n = 20$, for the whole dataset (i.e., $n = 2843$), the value of the curvature is small for all cases of $\alpha$, $\beta$ and $c$, demonstrating less effect of the prior in the presence of a large sample size.

**Table 4.** Values of the local curvature for the two classes $\Gamma_a$ and $\Gamma_g$ for the real data set AIDS.

| $n$ | $\begin{pmatrix} \alpha \\ \beta \end{pmatrix}$ | $c$ | $a = 0.5$ | | $a = 1$ | | $a = 2$ | |
|---|---|---|---|---|---|---|---|---|
| | | | $C_a^{\Gamma_a}$ | $C_a^{\Gamma_g}$ | $C_a^{\Gamma_a}$ | $C_a^{\Gamma_g}$ | $C_a^{\Gamma_a}$ | $C_a^{\Gamma_g}$ |
| 20 | $\begin{pmatrix} 0.5 \\ 0.5 \end{pmatrix}$ | 0.5 | 0.0001 | 0.0004 | 0.0003 | 0.0008 | 0.0006 | 0.0016 |
| | | 1 | 0 | 0 | 0 | 0 | 0 | 0 |
| | | 1.5 | 0.0006 | 0.0004 | 0.0012 | 0.0008 | 0.0025 | 0.0016 |
| | | 3 | 0.0174 | 0.0065 | 0.0348 | 0.0130 | 0.0697 | 0.0260 |
| | | 5 | 0.0876 | 0.0260 | 0.1752 | 0.0521 | 0.3504 | 0.1043 |
| | $\begin{pmatrix} 1 \\ 1 \end{pmatrix}$ | 0.5 | 0.0007 | 0.0014 | 0.0014 | 0.0028 | 0.0029 | 0.0057 |
| | | 1 | 0 | 0 | 0 | 0 | 0 | 0 |
| | | 1.5 | 0.0019 | 0.0014 | 0.0038 | 0.0028 | 0.0076 | 0.0057 |
| | | 3 | 0.0395 | 0.0229 | 0.0791 | 0.0458 | 0.1583 | 0.0916 |
| | | 5 | 0.1578 | 0.0916 | 0.3156 | 0.1832 | 0.6312 | 0.3665 |
| | $\begin{pmatrix} 1 \\ 3 \end{pmatrix}$ | 0.5 | 0.0049 | 0.0071 | 0.0099 | 0.0143 | 0.0198 | 0.0286 |
| | | 1 | 0 | 0 | 0 | 0 | 0 | 0 |
| | | 1.5 | 0.0075 | 0.0071 | 0.0150 | 0.0143 | 0.0301 | 0.0286 |
| | | 3 | 0.0995 | 0.1146 | 0.1991 | 0.2293 | 0.3982 | 0.4586 |
| | | 5 | 0.2799 | 0.4586 | 0.5599 | 0.9173 | 1.1198 | 1.8346 |
| | $\begin{pmatrix} 3 \\ 1 \end{pmatrix}$ | 0.5 | 0.0457 | 0.0319 | 0.0915 | 0.0638 | 0.1831 | 0.1277 |
| | | 1 | 0 | 0 | 0 | 0 | 0 | 0 |
| | | 1.5 | 0.0195 | 0.0319 | 0.0391 | 0.0638 | 0.0782 | 0.1277 |
| | | 3 | 0.0855 | 0.5111 | 0.1710 | 1.0223 | 0.3420 | 2.0446 |
| | | 5 | 0.1030 | 2.0446 | 0.2060 | 4.0892 | 0.4121 | 8.1784 |
| 2843 | $\begin{pmatrix} 0.5 \\ 0.5 \end{pmatrix}$ | 0.5 | $9 \times 10^{-7}$ | $2 \times 10^{-6}$ | $10^{-6}$ | $5 \times 10^{-6}$ | $3 \times 10^{-6}$ | $10^{-5}$ |
| | | 1 | 0 | 0 | 0 | 0 | 0 | 0 |
| | | 1.5 | $4 \times 10^{-6}$ | $2 \times 10^{-6}$ | $8 \times 10^{-6}$ | $5 \times 10^{-6}$ | $10^{-5}$ | $10^{-5}$ |
| | | 3 | 0.0001 | $4 \times 10^{-5}$ | 0.0003 | $8 \times 10^{-5}$ | 0.0006 | 0.0001 |
| | | 5 | 0.0009 | 0.0001 | 0.0019 | 0.0003 | 0.0038 | 0.0006 |
| | $\begin{pmatrix} 1 \\ 1 \end{pmatrix}$ | 0.5 | $4 \times 10^{-6}$ | $10^{-5}$ | $9 \times 10^{-6}$ | $2 \times 10^{-5}$ | $10^{-5}$ | $4 \times 10^{-5}$ |
| | | 1 | 0 | 0 | 0 | 0 | 0 | 0 |
| | | 1.5 | $10^{-5}$ | $10^{-5}$ | $3 \times 10^{-5}$ | $2 \times 10^{-5}$ | $6 \times 10^{-5}$ | $4 \times 10^{-5}$ |
| | | 3 | 0.0004 | 0.0001 | 0.0009 | 0.0003 | 0.0018 | 0.0006 |
| | | 5 | 0.0025 | 0.0006 | 0.0051 | 0.0013 | 0.0102 | 0.0027 |
| | $\begin{pmatrix} 1 \\ 3 \end{pmatrix}$ | 0.5 | 0.0005 | 0.0004 | 0.0010 | 0.0008 | 0.0021 | 0.0016 |
| | | 1 | 0 | 0 | 0 | 0 | 0 | 0 |
| | | 1.5 | 0.0002 | 0.0004 | 0.0004 | 0.0008 | 0.0008 | 0.0016 |
| | | 3 | 0.0002 | 0.0064 | 0.0004 | 0.0129 | 0.0009 | 0.0259 |
| | | 5 | $10^{-5}$ | 0.0259 | $3 \times 10^{-5}$ | 0.0518 | $7 \times 10^{-5}$ | 0.1037 |
| | $\begin{pmatrix} 3 \\ 1 \end{pmatrix}$ | 0.5 | $2 \times 10^{-5}$ | $5 \times 10^{-5}$ | $5 \times 10^{-5}$ | 0.0001 | 0.0001 | 0.0002 |
| | | 1 | 0 | 0 | 0 | 0 | 0 | 0 |
| | | 1.5 | $6 \times 10^{-5}$ | $5 \times 10^{-5}$ | 0.0001 | 0.0001 | 0.0002 | 0.0002 |
| | | 3 | 0.0014 | 0.0008 | 0.0029 | 0.0016 | 0.0058 | 0.0032 |
| | | 5 | 0.0054 | 0.0032 | 0.0108 | 0.0064 | 0.0216 | 0.0129 |

**Example 2 (Multinomial model).** *Suppose that* $\mathbf{x} = (x_1, x_2, \ldots, x_k)$ *is an observation from a multinomial distribution with parameters* $(N, (\theta_1, \ldots, \theta_k))$, *where* $\sum_{i=1}^{k} x_i = N$ *and* $\sum_{i=1}^{k} \theta_i = 1$. *Let the prior* $\pi_0(\theta_1, \ldots, \theta_k)$ *be Dirichlet*$(\alpha_1, \ldots, \alpha_k)$. *Then* $\pi_0(\theta_1, \ldots, \theta_k | \mathbf{x})$ *is Dirichlet*$(\alpha_1 + x_1, \ldots, \alpha_k + x_k)$.

Let $q(\theta_1, \ldots, \theta_k) \sim$ Dirichlet$(c\alpha_1, \ldots, c\alpha_k)$. We consider the observation $\mathbf{x} = (6, 4, 5, 5)$ generated from Multinomial$(20, (1/4, 1/4, 1/4, 1/4))$. As in Example 1, we use Monte Carlo approach to compute curvature values. Table 5 reports values of the curvature for different values of $\alpha_1, \ldots, \alpha_k$ and $c$. For the cases when $\alpha_1 = \alpha_2 = \alpha_3 = \alpha_4 = 1$ (uniform prior over $[0, 1]^4$) and $\alpha_1 = \alpha_2 = \alpha_3 = \alpha_4 = 0.5$ (Jeffreys' prior), the curvature values are prominently small.

**Table 5.** Values of the local curvature for two classes $\Gamma_a$ and $\Gamma_g$ for a sample generated from Mn(20,(1/4,1/4,1/4,1/4)).

| $\begin{pmatrix} \alpha_1 \\ \vdots \\ \alpha_4 \end{pmatrix}$ | $c$ | $a = 0.5$ | | $a = 1$ | | $a = 2$ | |
|---|---|---|---|---|---|---|---|
| | | $C_a^{\Gamma_a}$ | $C_a^{\Gamma_g}$ | $C_a^{\Gamma_a}$ | $C_a^{\Gamma_g}$ | $C_a^{\Gamma_a}$ | $C_a^{\Gamma_g}$ |
| $\begin{pmatrix} 0.25 \\ 0.25 \\ 0.25 \\ 0.25 \end{pmatrix}$ | 0.5 | $2 \times 10^{-5}$ | 0.0006 | $5 \times 10^{-5}$ | 0.0012 | 0.0001 | 0.0024 |
| | 1 | 0 | 0 | 0 | 0 | 0 | 0 |
| | 1.5 | 0.0031 | 0.0006 | 0.0062 | 0.0012 | 0.0124 | 0.0024 |
| | 3 | 0.5285 | 0.0097 | 1.0570 | 0.0195 | 2.1141 | 0.0390 |
| | 5 | 8.4050 | 0.0301 | 16.816 | 0.0780 | 33.632 | 0.1560 |
| $\begin{pmatrix} 0.5 \\ 0.5 \\ 0.5 \\ 0.5 \end{pmatrix}$ | 0.5 | 0.0001 | 0.0021 | 0.0003 | 0.0043 | 0.0004 | 0.0087 |
| | 1 | 0 | 0 | 0 | 0 | 0 | 0 |
| | 1.5 | 0.0080 | 0.0021 | 0.0161 | 0.0043 | 0.0323 | 0.0087 |
| | 3 | 0.7706 | 0.0349 | 1.5413 | 0.0699 | 3.0826 | 0.1398 |
| | 5 | 8.0246 | 0.1398 | 16.049 | 0.2797 | 32.098 | 0.5595 |
| $\begin{pmatrix} 1 \\ 1 \\ 1 \\ 1 \end{pmatrix}$ | 0.5 | 0.0008 | 0.0071 | 0.0017 | 0.0142 | 0.0035 | 0.0284 |
| | 1 | 0 | 0 | 0 | 0 | 0 | 0 |
| | 1.5 | 0.0185 | 0.0071 | 0.0370 | 0.0142 | 0.0741 | 0.0284 |
| | 3 | 0.9799 | 0.1137 | 1.9598 | 0.2274 | 3.9196 | 0.4549 |
| | 5 | 6.7661 | 0.4549 | 13.532 | 0.9098 | 27.064 | 1.8197 |
| $\begin{pmatrix} 2 \\ 1 \\ 1 \\ 1 \end{pmatrix}$ | 0.5 | 0.0018 | 0.0120 | 0.0037 | 0.0240 | 0.0074 | 0.0480 |
| | 1 | 0 | 0 | 0 | 0 | 0 | 0 |
| | 1.5 | 0.0270 | 0.0120 | 0.0540 | 0.0240 | 0.1081 | 0.0480 |
| | 3 | 1.1052 | 0.1923 | 2.2104 | 0.3847 | 4.4209 | 0.7695 |
| | 5 | 6.3984 | 0.7695 | 12.796 | 1.5390 | 25.593 | 3.0780 |

**Example 3 (Location normal model).** *Suppose that* $\mathbf{x} = (x_1, x_2, \ldots, x_n)$ *is a sample from* $N(\theta, 1)$ *distribution with* $\theta \in \mathbb{R}^1$. *Let the prior* $\pi_0(\theta)$ *of* $\theta$ *be* $N(\theta_0, \sigma_0^2)$. *Then*

$$\pi_0(\theta | \mathbf{x}) \sim \mathcal{N}\left(\mu_{\mathbf{x}}, \sigma_{\mathbf{x}}^2\right), \tag{20}$$

$$\mu_{\mathbf{x}} = \left(\frac{\theta_0}{\sigma_0^2} + n\bar{\mathbf{x}}\right)\left(\frac{1}{\sigma_0^2} + n\right)^{-1} \text{ and } \sigma_{\mathbf{x}}^2 = \left(\frac{1}{\sigma_0^2} + n\right)^{-1}.$$

Let $q(\theta) \sim \mathcal{N}(c\theta_0, \sigma_0^2)$, $c > 0$. Due to some interesting theoretical properties in this example, we present the exact formulas of the curvature for class (1) and class (2). We have

$$\frac{q(\theta)}{\pi_0(\theta)} = \exp\left\{\frac{\theta_0\theta(c-1) + 0.5\theta_0^2(1-c^2)}{\sigma_0^2}\right\}.$$

Therefore, for the class (1), we have

$$
\begin{aligned}
Var_{\pi_0(\theta|\mathbf{x})}\left[\frac{q(\theta)}{\pi_0(\theta)}\right] &= E_{\pi_0(\theta|\mathbf{x})}\left[\left(\frac{q(\theta)}{\pi_0(\theta)}\right)^2\right] - \left(E_{\pi_0(\theta|\mathbf{x})}\left[\frac{q(\theta)}{\pi_0(\theta)}\right]\right)^2 \\
&= \exp\left\{\frac{\theta_0^2(1-c^2)}{\sigma_0^2}\right\}\left[M_{\pi_0(\theta|\mathbf{x})}\left(\frac{2\theta_0(c-1)}{\sigma_0^2}\right) - \right. \\
&\qquad \left. \left(M_{\pi_0(\theta|\mathbf{x})}\left(\frac{\theta_0(c-1)}{\sigma_0^2}\right)\right)^2\right],
\end{aligned}
$$

where $M_{\pi_0(\theta|\mathbf{x})}(t)$ is the moment generating function with respect to the density $\pi_0(\theta|\mathbf{x})$. Thus, $Var_{\pi_0(\theta|\mathbf{x})}\left[\frac{q(\theta)}{\pi_0(\theta)}\right]$ is equal to

$$
\begin{aligned}
&\exp\left\{\frac{\theta_0^2(1-c^2)}{\sigma_0^2}\right\}\left[\exp\left\{\frac{2\theta_0(c-1)\mu_{\mathbf{x}}}{\sigma_0^2} + \frac{2\theta_0^2(c-1)^2\sigma_{\mathbf{x}}^2}{\sigma_0^4}\right\} - \exp\left\{\frac{2\theta_0(c-1)\mu_{\mathbf{x}}}{\sigma_0^2} + \right.\right. \\
&\qquad \left.\left.\frac{\theta_0^2(c-1)^2\sigma_{\mathbf{x}}^2}{\sigma_0^4}\right\}\right] \\
&= \exp\left\{\frac{\theta_0^2(1-c^2)}{\sigma_0^2}\right\}\exp\left\{\frac{2\theta_0(c-1)\mu_{\mathbf{x}}}{\sigma_0^2}\right\}\exp\left\{\frac{\theta_0^2(c-1)^2\sigma_{\mathbf{x}}^2}{\sigma_0^4}\right\} \\
&\qquad \times \left[\exp\left\{\frac{\theta_0^2(c-1)^2\sigma_{\mathbf{x}}^2}{\sigma_0^4}\right\} - 1\right].
\end{aligned}
$$

On the other hand, for the geometric contaminated class, we have

$$
\ln\left(\frac{q(\theta)}{\pi_0(\theta)}\right) = \frac{\theta_0\theta(c-1) + 0.5\theta_0^2(1-c^2)}{\sigma_0^2}.
$$

Thus, by (20), we get

$$
\begin{aligned}
Var_{\pi_0(\theta|\mathbf{x})}\left[\ln\left(\frac{q(\theta)}{\pi_0(\theta)}\right)\right] &= \frac{\theta_0^2(c-1)^2}{\sigma_0^4}Var_{\pi_0(\theta|\mathbf{x})}[\theta] \\
&= \frac{\theta_0^2(c-1)^2}{\sigma_0^4}\sigma_{\mathbf{x}}^2 \\
&= \frac{\theta_0^2(c-1)^2}{\sigma_0^4}\left(\frac{1}{\sigma_0^2} + n\right)^{-1}.
\end{aligned}
\tag{21}
$$

Interestingly, from (21), $Var_{\pi_0(\theta|\mathbf{x})}\left[\ln\left(\frac{q(\theta)}{\pi_0(\theta)}\right)\right]$ depends on the sample only through its size $n$. For fixed values of $\theta_0$ and $c$, as $n \to \infty$ or $\sigma_0 \to \infty$, $Var_{\pi_0(\theta|\mathbf{x})}\left[\ln\left(\frac{q(\theta)}{\pi_0(\theta)}\right)\right] \to 0$, which indicates robustness. Also, for fixed values of $\sigma_0$ and $n$, as $\theta_0 \to \infty$ or $c \to \infty$, $Var_{\pi_0(\theta|\mathbf{x})}\left[\ln\left(\frac{q(\theta)}{\pi_0(\theta)}\right)\right] \to \infty$ and no robustness will be found.

Now we consider a numerical example by generating a sample of size $n = 20$ from $N(4, 1)$ distribution. We obtain

$$
\begin{aligned}
\mathbf{x} = (&3.37, 4.18, 3.16, 5.59, 4.32, 3.17, 4.48, 4.73, 4.57, 3.69, 5.51, 4.38, 3.37, \\
&1.78, 5.12, 3.95, 3.98, 4.94, 4.82, 4.59)
\end{aligned}
$$

(with $t = \bar{x} = 4.1905$). Table 6 reports the values of the curvature for different values of $\theta_0, \sigma_0$ and $c$.

**Table 6.** Values of the local curvature for two classes $\Gamma_a$ and $\Gamma_g$ for a sample generated from N(4,1).

| $\begin{pmatrix}\theta_0\\\sigma_0^2\end{pmatrix}$ | $c$ | $a = 0.5$ | | $a = 1$ | | $a = 2$ | |
|---|---|---|---|---|---|---|---|
| | | $C_a^{\Gamma_a}$ | $C_a^{\Gamma_g}$ | $C_a^{\Gamma_a}$ | $C_a^{\Gamma_g}$ | $C_a^{\Gamma_a}$ | $C_a^{\Gamma_g}$ |
| $\begin{pmatrix}0.1\\0.1\end{pmatrix}$ | 0.5 | 0.0001 | 0.0059 | 0.0002 | 0.0119 | 0.0004 | 0.0238 |
| | 1 | 0 | 0 | 0 | 0 | 0 | 0 |
| | 1.5 | 0.2908 | 0.0059 | 0.5816 | 0.0119 | 1.1633 | 0.0238 |
| | 3 | 498,033.7 | 0.0953 | 996,067.4 | 0.1907 | 1,992,135 | 0.3814 |
| | 5 | $8 \times 10^{12}$ | 0.3814 | $10^{13}$ | 0.7629 | $3 \times 10^{13}$ | 1.5258 |
| $\begin{pmatrix}0.5\\1\end{pmatrix}$ | 0.5 | 0.0002 | 0.0014 | 0.0004 | 0.0029 | 0.0009 | 0.0059 |
| | 1 | 0 | 0 | 0 | 0 | 0 | 0 |
| | 1.5 | 0.0081 | 0.0014 | 0.0162 | 0.0029 | 0.0325 | 0.0059 |
| | 3 | 10.629 | 0.0238 | 21.258 | 0.0476 | 42.517 | 0.0953 |
| | 5 | 2964.9 | 0.0935 | 2929.8 | 0.1907 | 11,859.7 | 0.3814 |
| $\begin{pmatrix}0.5\\5\end{pmatrix}$ | 0.5 | $4 \times 10^{-5}$ | $5 \times 10^{-5}$ | $8 \times 10^{-5}$ | 0.0001 | 0.0001 | 0.0002 |
| | 1 | 0 | 0 | 0 | 0 | 0 | 0 |
| | 1.5 | $8 \times 10^{-5}$ | $5 \times 10^{-5}$ | 0.0001 | 0.0001 | 0.0003 | 0.0002 |
| | 3 | 0.0031 | 0.0009 | 0.0063 | 0.0019 | 0.0127 | 0.0038 |
| | 5 | 0.0288 | 0.0038 | 0.0576 | 0.0076 | 0.1152 | 0.0152 |
| $\begin{pmatrix}4\\5\end{pmatrix}$ | 0.5 | 0.0001 | 0.0038 | 0.0029 | 0.0076 | 0.0059 | 0.0152 |
| | 1 | 0 | 0 | 0 | 0 | 0 | 0 |
| | 1.5 | 0.0020 | 0.0038 | 0.0040 | 0.0076 | 0.0080 | 0.0152 |
| | 3 | $3 \times 10^{-7}$ | 0.0610 | $7 \times 10^{-7}$ | 0.1220 | $10^{-6}$ | 0.2441 |
| | 5 | $9 \times 10^{-23}$ | 0.2441 | $10^{-22}$ | 0.4882 | $3 \times 10^{-22}$ | 0.9765 |

Clearly, for large values of $\sigma_0^2$, the value of the curvature is small, which is an indication of robustness. For instance, for $\mu_0 = 0.5$ in Table 6, that value of the curvature when $\sigma_0^2 = 5$ is much smaller than the value of the curvature when $\sigma_0^2 = 1$.

## 5. Conclusions

Measuring Bayesian robustness of two classes of contaminated priors is studied. The approach is based on computing the curvature of Rényi divergence between posterior distributions. Two different proofs are given for the results. The first one is general and depends on a direct derivation of the curvatures. The second one uses the connection between $(h, \phi)$ divergence and $\phi$ divergence. The derived results do not require specifying values for $\epsilon$ and its computation is straightforward. Examples illustrating the approach are considered. Finally, it is possible to extend the results in this paper to other divergences. See, for instance, Liese and Vajda (1982) [24]. We leave this direction for future work.

**Author Contributions:** All authors have contributed equally on this work. All authors have read and agreed to the published version of the manuscript.

**Funding:** This research received no external funding.

**Institutional Review Board Statement:** Not applicable.

**Informed Consent Statement:** Not applicable.

**Data Availability Statement:** Not available.

**Acknowledgments:** The authors thank the Editor, the Associate Editor and anonymous referees for their important and constructive comments that led to significant improvement of the paper. In particular, the connection between $(h, \phi)$ divergence and $\phi$ divergence is highly appreciated.

**Conflicts of Interest:** The authors declare no conflict of interest.

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
