# Peer review of "Measuring Bayesian Robustness Using Rényi Divergence"

_stats, doi:10.3390/stats4020018_

Round 1
Reviewer 1 Report
In this paper the authors present how to measure Bayesian robustness using Renyi divergence. The article is very interesting and here are some suggestions that could make it easier for the reader to better understand the usefulness of your proposal.
As the authors comment, Renyi divergence includes Kullbak-Leibler divergence as a particular case when a = 1. It would be interesting for the reader if the authors explain in which cases the divergence of order other than 1 could be relevant. In the examples shown, the authors propose only three possible values of a = (0.5, 1, and 2). Would it be interesting to study the divergence for an order greater than 2? In what cases?
The authors derive the expression from the local curvature of the Renyi divergence. In the examples they only calculate the Renyi divergence and the calibrated value p example 1. Why only for example 1? Is it only due to reduce the size of the paper or is there some limitation to calculate these values in the other two examples shown?
Page 15, the autors explain the relationship between Table 1 and Tables 2 and 3 but the example is for alpha=1 and beta =3 (not 1)
Page 15. The sentence "Clearly, when c=1 the curvature values are 0" should be written in the first example as it is general for all cases.
Page 18. The sentence "for the cases... the curvature values are prominently small" is not so clear in Table 5. There are several examples in which the local curvature is smaller for the (0.25,0.25,0.25,0.25) or (2,1,1,1). How can the authors explain this fact?
The authors should expand the conclusions indicating when the use of Renyi divergence may be more useful than other divergences already studied in the literature.
Author Response
Kindly see attached file.

Reviewer 2 Report
See the attached review report.

Author Response
Kindly see attached file.

Reviewer 3 Report
Please see attachment. Thanks.

Author Response
Kindly see attached file.

Reviewer 4 Report
The comments are in the pdf file

Author Response
Kindly see attached file.

Round 2
Reviewer 3 Report
Please see attachment

Author Response
Thank you! The required codes are attached. They are used to generate all the tables in Example 1. The code can be modified easily for other examples.
Reviewer 4 Report
The paper can be accepted
Author Response
Thanks you!